# Genetic and structural insights into the functional importance of the conserved gly-met-rich C-terminal tails in bacterial chaperonins

C. M. Santosh Kumar [1] ✉, Aisha M. Mai[1], Shekhar C. Mande [2,3] & Peter A. Lund [1]

*E. coli* chaperonin GroEL forms nano-cages for protein folding. Although the chaperonin-mediated protein folding mechanism is well understood, the role of the conserved glycine and methionine-rich carboxy-terminal residues remains unclear. Bacteria with multiple chaperonins always retain at least one paralogue having the gly-met-rich C-terminus, indicating an essential conserved function. Here, we observed a stronger selection pressure on the paralogues with gly-met-rich C-termini, consistent with their ancestral functional importance. *E. coli* GroEL variants having mutations in their C-termini failed to functionally replace GroEL, suggesting the functional significance of the gly-met-rich C-termini. Further, our structural modelling and normal mode analysis showed that the C-terminal region shuttles between two cavity-specific conformations that correlate with the client-protein-binding apical domains, supporting C-termini's role in client protein encapsulation. Therefore, employing phylogenetic, genetic, and structural tools, we demonstrate that the gly-met-rich C-termini are functionally significant in chaperonin-mediated protein folding function. Owing to the pathogenic roles of the chaperonins having non-canonical C-termini, future investigations on the client protein selectivity will enable understanding the disease-specific client protein folding pathways and treatment options.

The bacterial chaperonin GroEL, with its cofactor GroES, constitutes an essential molecular machine for cellular protein folding in *E. coli*, and homologues exist in nearly all organisms[1–4]. GroEL forms a double toroidal tetradecamer of ~900 kD, which has two solvent-filled cavities, one in each heptameric ring[5,6]. Extensive investigations have dissected the structure-function relationships of GroEL's domains[7–9] and the movements that they undergo during the ATP-driven cycles of client protein binding, encapsulation, folding, and release[3,10]. Although the GroEL cavities enable client protein folding, the precise role of the cavity is still debated[1,3,4,11]. Some models proposed that the cavities play an active role in enhancing the folding of the encapsulated client protein[4,7,12], while others propose that the cavity acts as a passive receptacle[13,14]. Although chaperonin-client interactions have been attributed to the hydrophobic patches in the cavity rim[1], a complete understanding of the mechanism by which client proteins are internalized and expelled post-folding is lacking.

The terminal 23 amino-acid residues of GroEL, which end with a highly glycine-rich peptide ((GGM)$_4$M), are not resolved in crystal structures of the GroEL complex, suggesting that they are either disordered or exhibit a range of possible structures even when the protein is crystallised. Structural models of *E. coli* (https://alphafold.ebi.ac.uk/entry/P0A6F5) and *Bacillus subtills* (https://alphafold.ebi.ac.uk/entry/P28598) GroELs generated by the AlphaFold show extended carboxy termini. These models considered monomeric forms, not the energy-efficient oligomeric single-ring or bullet forms. The extended conformation of the carboxy termini would cause steric hindrance when monomers assemble into single or double ring conformations. In this paper, we refer to this terminal 23-amino-acid residue part of the protein as the carboxy terminal segments (CTSs). Crystal structures show a possible opening at the bottom of the cavity that could suggest that the two cavities are connected[5,6]. However, cryo-electron microscopy and SANS both demonstrated elec-

[1]Institute of Microbiology and Infection, School of Biosciences, University of Birmingham, Birmingham, UK. [2]National Centre for Cell Science, Pune, India. [3]Present address: Bioinformatics Centre, Savitribai Phule Pune University, Pune, India. ✉e-mail: s.k.cm@bham.ac.uk

tron density at the base of the cavity, which would prevent client protein movement between the two heptameric rings, and it is generally agreed that this density arises from these CTSs[15,16]. This region becomes more ordered when ATP is bound[17]. Cryo-EM structures of a complex representing an early intermediate in protein encapsulation showed significant association between this C-terminal region and the bound client protein, implying that this region may have a role in the internalization of the client proteins[18]. Moreover, analysis of the folding of green fluorescent protein (GFP) in a single ring version of GroEL lacking the CTS suggested that denatured but not folded GFP could leak out from the bottom of the GroEL cage[19], consistent with structural data showing that the CTS normally forms a barrier between the two cavities. Thus, the CTS may have multiple roles in the GroEL folding cycle, both in binding and internalizing client proteins and in preventing their leakage from the equatorial ends of the heptameric rings.

Genetic and biochemical studies have also analysed the role of this region. Exchanging CTSs between chaperonin homologues from organisms with different optimal growth temperatures has demonstrated that the CTS could act as a thermometer for group I[20] and group II[21,22] chaperonins. In contrast to the above findings, genetic studies in *E. coli* that demonstrated that GroEL can function without the terminal sixteen[23] or twenty-seven residues[24,25] suggest that the CTS is dispensable for chaperonin function. GroEL with a C-terminal deletion of 27 amino-acids was reported to support the growth of a *groEL Ts* mutant, bacteriophage morphogenesis, and growth of a *groEL* deletion strain, though this required high levels of the truncated GroEL[24]. GroEL with a further single amino-acid deletion failed to complement and was shown to be assembly deficient[14,25].

Biochemical studies on a range of purified truncated GroEL molecules showed that deletion of the 23 C-terminal amino-acids caused reduced ATPase activity and chaperone activity with a range of client proteins. However, shorter deletions of seven or seventeen amino acids had very little effect on these properties[26]. Furthermore, GroEL variants with extended C-terminal segments showed client and extension size-specific variations in folding rates. These observations have been attributed to changes in either the cavity volume[7,12] or ATPase activity[13], suggesting that the GroEL C-terminus plays an important role in modulating the chaperonin reaction cycle[12,13]. Recent structural and biochemical studies have suggested that the CTS might form a sieve between the cavities[19,27] and interact with client proteins, favouring their unfolding[18,28,29]. The fact that chaperonins tagged at the carboxy terminus with a hexa-histidine tag bind to Ni-NTA resin, as reported in several studies aimed at purifying chaperonin tetradecamers[30] or the complex with bound co-chaperonins[31], implies that the CTSs can reach close to the cavity rim, at least in the GroES-free *trans* ring, as this would be required to enable the (His)$_6$ tag to bind to the Ni-NTA resin. Molecular dynamic simulations[32] and predictive structural studies also show that the CTSs might reach the cavity rim and sometimes extend out of the cavity. Taken together, these studies suggest that the CTS plays an important role in GroEL-mediated protein folding by binding to the client proteins and exhibiting multiple conformational states. However, how the CTS contributes to GroEL's function remains unclear. Therefore, we reevaluated the function of the CTS.

To understand CTSs' functional relevance, we examined their conservation. The CTS of type I and type II chaperonins were observed to be highly conserved, with the variation usually in the number of repeating GGM motifs[33,34]. However, some CTSs in the bacterial chaperonins, especially multiple chaperonin genes in the same species, were observed to deviate from the gly-met-rich composition. Examples of bacteria encoding multiple chaperonin genes include several proteobacteria (with *Bradyrhizobium japonicum* encoding seven chaperonin genes), Cyanobacteria, and Actinobacteria[2,3]. The proteobacterial and cyanobacterial chaperonin CTSs were either gly-met-rich or pattern-free, while the actinobacterial CTS exhibited distinct divergence in their composition, encoding histidine-rich, glutamine-rich, or pattern-less CTSs[2,35,36]. Interestingly, the paralogues with a gly-met rich CTS were often observed to be essential, while others were often dispensable[2,37], suggesting an essential role for gly-met-rich CTS.

Here, we report investigations into the function of the chaperonin CTS. A phylogenetic analysis of 325 chaperonin sequences demonstrates that the CTS is likely to be a functionally significant part of chaperonins. To test this, we constructed and analysed a series of *E. coli* GroEL CTS variants and used normal mode analysis to study the possible conformations of the CTSs in the *E. coli* GroEL oligomer. Our results suggest that CTSs are likely to assume cavity-specific conformations that coordinate with the motions of the apical domains, consistent with the proposed roles for CTSs in client-recognition and encapsulation.

## Results

### Phylogenetic analysis suggests functional relevance of chaperonin CTSs

About 30% of fully sequenced bacterial genomes encode multiple chaperonin genes[2,3,38], with the majority of such examples belonging to actinobacteria. In the actinobacteria, one chaperonin paralogue has the gly-met-rich CTS, while the other paralogues have CTSs with different amino acid sequences, suggesting that the deviation in CTS composition is likely to be evolutionarily driven with an organism-specific relevance. Therefore, we examined the diversity of the chaperonins among 325 actinobacterial chaperonin paralogues. Chaperonin polypeptide sequences were retrieved from the UniProt database, and their meta data was sourced from literature (Supplementary data 1). The phylogenetic relationships of the full-length chaperonin polypeptide sequences were inferred using a neighbour-joining algorithm. In the resulting polynomial-time phylogenetic tree, the chaperonin sequences are grouped into three clades (Fig. 1a and S1). Interestingly, this classification is largely based on the chaperonins' CTS composition; the three clades largely contained chaperonins having a) the classical gly-met-rich CTS, b) Histidine-rich CTS or c) pattern-free CTS (Fig. S1). The chaperonins in the gly-met-rich group exhibit shorter branch lengths than the his-rich group (Fig. S1), showing higher conservation of the former. This correlates with the observation that the chaperonins in the gly-met-rich group are largely essential, as this would lead to higher selection pressure and, hence, lower divergence[39,40]. The chaperonins in the third group exhibit greater divergence and may represent independent acquisitions by horizontal gene transfer, as most actinobacteria only have two chaperonin genes[41] (Fig. S1). Our analysis is consistent with the hypothesis of an ancient chaperonin gene duplication in an ancestral actinobacterium followed by functional specialisation[3,39].

To test if the principal variation among the analysed chaperonin sequences was in their CTS regions, we used the full-length sequences of the chaperonins and calculated the residue-specific divergence rates, using the rate4site algorithm[41] and EMBOSS Cons. The resultant rates were mapped onto *E. coli* GroEL (Fig. 1b). As expected, the substrate binding helices H and I, residues involved in ATP binding hydrolysis (D52, D87 and D393), and a majority of the intermediate domains and inter-domain boundaries showed higher conservation with a divergence score close to zero, while the loop between helices N and O and flexible regions of apical domain showed lower conservation. Interestingly, the CTS showed high divergence compared to the rest of the sequence (Fig. 1b). EMBOSS Cons, which derives consensus sequence from an MSA, also showed CTS divergence (Fig. S2), in agreement with the Rate4Site analysis. This is consistent with a significant contribution of the CTS to the divergence in the chaperonin sequences, as observed in the sequence logo (Fig. 1c). To examine this further, we compared the residue-specific distance matrices of the full-length sequences with the CTS regions. The CTS regions showed higher overall divergence than the full-length sequences (Supplementary Data 2), indicating that much of the divergence in the chaperonins was contributed by the CTS regions.

We further analyzed the diverse actinobacterial CTS by plotting their average hydrophobicity (GRAVY) as a function of their charge (pI). Most of the CTS fell into one of three groups. The first group constitutes the gly-met-rich CTSs; these are tightly clustered with a mean pI around 4 and a GRAVY score between about 0.7 and -1.5. The second group constitutes charged-residue CTSs, which are relatively more dispersed with pI between 3 and 8 and a lower GRAVY score. The third group constitutes the pattern-free CTS,

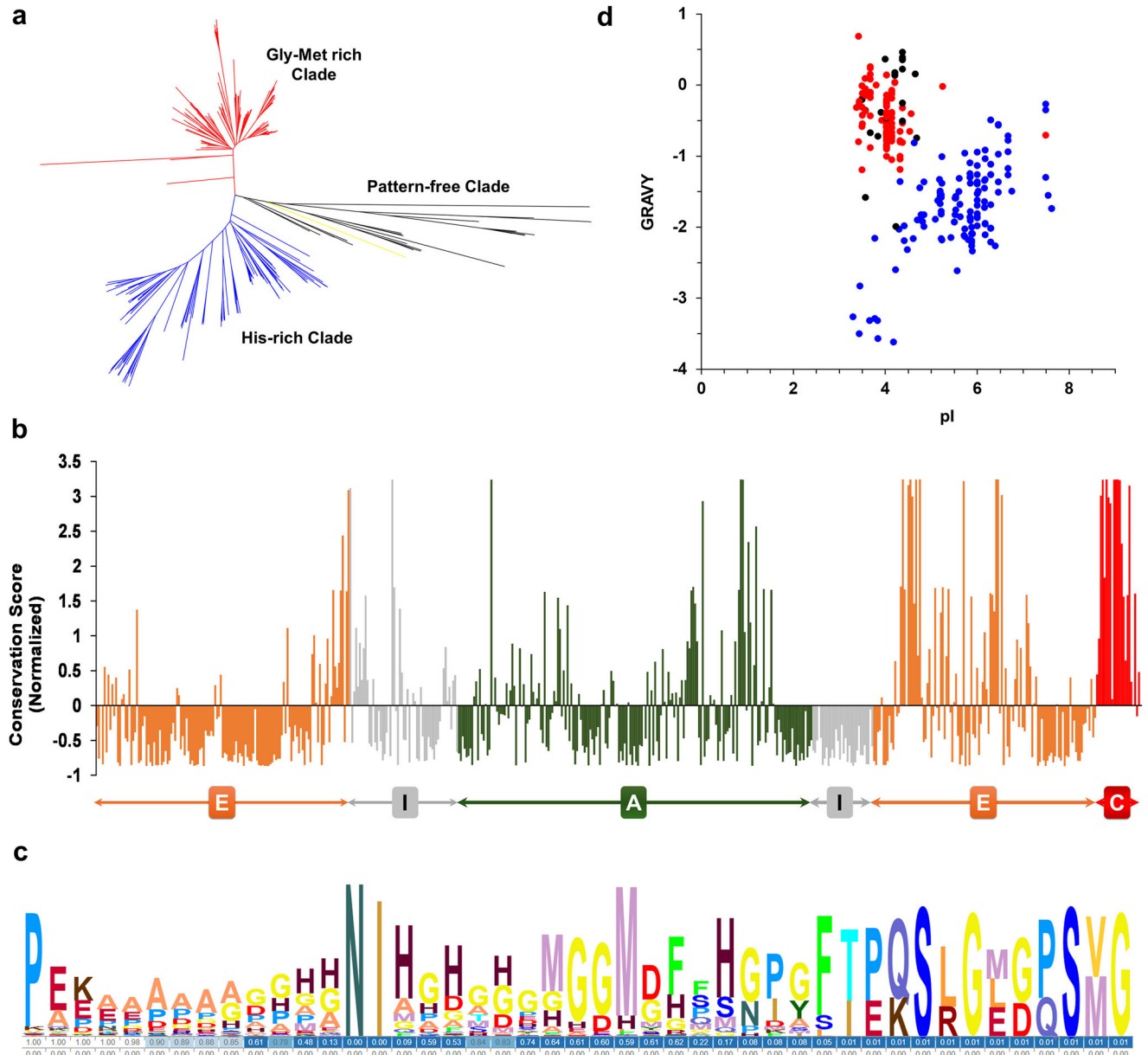

**Fig. 1 | Phylogenetic analysis of actinobacterial chaperonins. a** Phylogenetic relationships between 325 actinobacterial chaperonin sequences were inferred using a neighbourhood-joining algorithm and presented as a phylogenetic tree. The phylogenetic tree shows three clades. The yellow-coloured branch represents the *E. coli* GroEL sequence, which was included in the alignment. **b** Site specific conservation rates were calculated using ConSurf's Rate4Site algorithm from the multiple sequence analysis and mapped onto *E. coli* GroEL sequence. The graph represents the normalized conservation scores (lowest score represents conservation, with a standard deviation of one) as a function of GroEL's primary sequence.

The domain regions of the equatorial (E), intermediate (I) and apical (A) domains and the CTS region (C) are indicated. **c** Sequence Logo depicting the diversity in the CTS region of the actinobacterial chaperonins, starting with the conserved Proline. **d** Scatter plot showing hydrophobicity of the chaperonin CTSs (GRAVY Scores) as a function of their average charges (pI). Each dot represents one CTS and the spots are colour coded in the same way as the three branches in the phylogenetic tree. A detailed phylogenetic tree with taxon names and node ages is presented in the supplementary information.

and these have variable hydrophobicity (Fig. 1d). The relative clustering of the first and second groups confirms that, as expected, the CTS's physicochemical properties largely agree with their phylogenetic relationships.

## The GroEL carboxy terminus has an essential role in chaperonin function

As the observations above suggested that the CTS may be important in chaperonin function, we tested the ability of a GroEL variant that lacks the thirteen-residue C-terminus (GroEL$\Delta$C$_{13}$) to functionally replace full-length GroEL. We tested this using a *groEL* deletion strain, AI90[42], wherein the chromosomal *groEL* gene is replaced with a *kan$^R$* cassette and *groEL* is

expressed from a lactose-inducible $P_{lac}$ promoter in the shelter plasmid, pTGroEL7 (Cam$^r$, p15A Ori). Therefore, AI90 can lose the shelter plasmid, and become chloramphenicol sensitive only if the vector-borne chaperonin variants are functional. AI90 was transformed with plasmids expressing either GroEL or GroEL$\Delta$C$_{13}$ genes from a lactose-regulated promoter in the plasmid pTrc99A, and from an arabinose-regulated promoter in the plasmid pBAD24. The chloramphenicol sensitivity of the respective strains was assessed by serial dilution following the induction of the promoters regulating the vector-borne *groEL* variants. We observed that only the strains producing the full-length GroEL were able to lose the shelter plasmid and become chloramphenicol sensitive (Fig. 2a). As this method involved screening for the

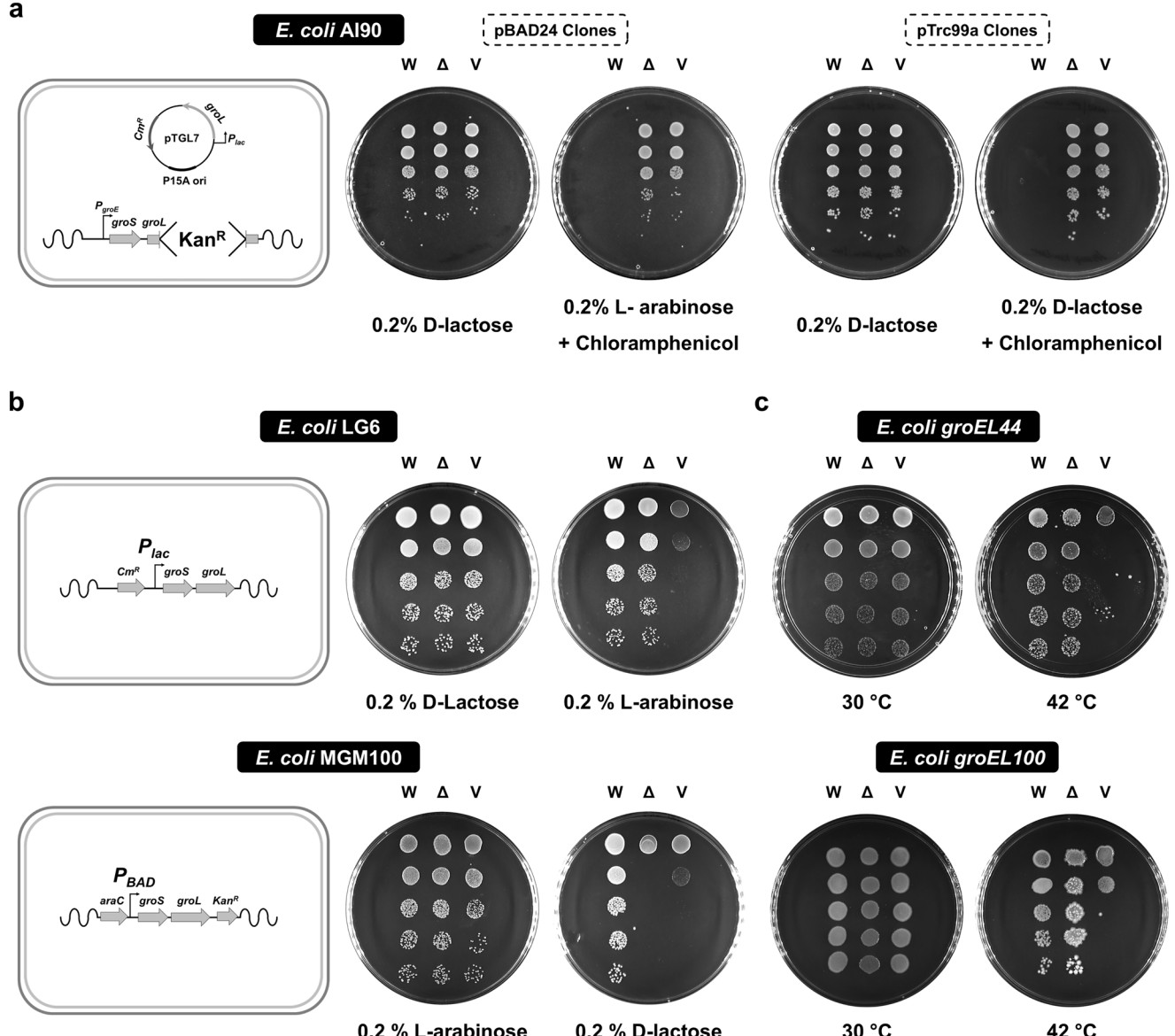

**Fig. 2 | The carboxy terminus is essential for full GroEL function.** The ability of *groEL* lacking the 13 c-terminal residues to functionally replace *groEL* was assessed in (**a**) the *groEL* deletion strain AI90; (**b**) the conditional expression strains LG6 and MGM100 and (**c**) temperature-sensitive strains *groEL44* and *groEL100*. Serially diluted cultures of the indicated *E. coli* strains expressing either the wildtype (W) or the 13 residue carboxy termini lacking *groEL* variant GroELΔC₁₃ (Δ) from a plasmid, or the vector only control (V), were spotted on LB agar plates supplemented as indicated. All the plates in **a** and **b** were incubated at 30 °C, while the plates in **c** were incubated at the indicated temperatures. The first and second plates in **b** and

**c** represent permissive and restrictive growth conditions, respectively. Relevant genetic features of the strains in **a** and **b** are depicted schematically. In AI90, *groEL* has been replaced by the *kan*^R cassette while a functional copy, regulated by lactose inducible *P_lac* promoter, is provided on a shelter plasmid pTGroEL7 (p15A, *cam*^R). The ability of the incoming indicated *groEL* variants to allow loss of pTGroEL7 was assessed. In LG6 and MGM100, the chromosomal copies of the bicistronic *groE* operon are controlled by lactose-inducible *P_lac* and arabinose-inducible *P_BAD* promoters, respectively.

loss of resistance, and hence rare events would be hard to detect, we further attempted to P1 transduce the D*groEL::kan*^R marker from AI90 into *E. coli* MG1655 harbouring the plasmids encoding the same *groEL* CTS variants. The strain producing GroEL_wt formed *kan*^R colonies at high frequency, but those producing GroELΔC₁₃ failed to produce any *kan*^R colonies.

As this result was different to that reported previously[24,25], we further assessed the function of GroELΔC₁₃ in two *groEL* depletion strains, *E. coli* LG6[14] and MGM100[43], wherein the chromosomal *groESL* operon is regulated by *P_lac* and *P_BAD* promoters, respectively. GroELΔC₁₃ was expressed in *E. coli* LG6 from pBAD24 and in MGM100 from pTrc99A. Growth of the respective strains was assessed by serial dilution following repression of the promoters regulating the chromosomal *groESL* operon. GroELΔC₁₃ rescued the growth of LG6, but not of MGM100 (Fig. 2b). The *P_BAD* promoter

that controls the expression of the *groESL* operon in MGM100 is known to be more tightly repressed than the *P_lac* promoter in LG6[43,44], so this difference probably arises from weak expression of the chromosomal *groEL* gene in LG6, but not in MGM100. We confirmed this using reverse transcriptase PCR (Fig. S3).

To further assess if GroELΔC₁₃ can independently function as a chaperonin, we investigated its ability to complement two strains that carry temperature sensitive (*Ts*) *groEL* alleles, SV2 (*groEL44*[45]) and A7579 (*groEL100*[46]), which encode *Ts* chaperonin variants, GroEL E191G and S201F, respectively. GroELΔC₁₃ was able to rescue the *Ts* phenotypes of these strains at their non-permissive temperature (Fig. 2c).

Taken together, these observations show that GroELΔC₁₃ retains partial function, which can enable growth when sufficient full-length GroEL

is present in the cell, even if it is at lower levels (as in LG6) or itself only partially functional (as in SV2 and A7579). However, when GroEL is present at too low levels (as in MGM100) or is absent (as in AI90), GroELΔC[13] is not able to support growth. We therefore predicted that it would be impossible to delete the *groEL* gene from a strain only expressing GroELΔC[13].

### The CTSs exhibit correlated motions with the apical domains during the chaperonin cycle

Having established that the CTS regions are essential for chaperonin function under the conditions of our assay, we wished to study the structure and movement of the CTSs within the GroEL tetradecamer. In lieu of detailed structural information[5,6], we used in silico approaches to study CTS dynamics. The suitability of these methods for studying GroEL dynamics was assessed by checking for their consistency with experimental findings (Supplementary Results). Structural models of GroEL protomers (one from each ring) with (GGM)[4]M carboxy terminus were generated using Modeller and COOT. Models with the lowest energy were selected and superposed to obtain GroEL tetradecamers (Fig. 3a) with a *cis* ring in the relaxed (R") conformational state) and a *trans* ring in the tight (T) conformational state) ring (Fig. 3a). In these models, the CTSs assume cavity-specific conformations (Fig. 3a, b), similar to those seen in cryo-EM structures[47], filling the void at the base of the cavities (Fig. S4). Since GroEL shuttles between *T* and *R"* states during its functional cycle[48], we employed Normal Mode Analysis to generate successive states of structural intermediates to map the path taken to traverse between *T* and *R"* states. Large dynamics of the CTSs that were observed in the protomer (that are −4–10 times stronger than averaged atomic displacements in the rest of molecule (Fig. S5)), were both reduced (Fig. S6) and synchronized (Fig. S7) in the heptameric ring. In other words, the ring arrangement of protomers in GroEL restricts the dynamic movements of the CTSs. Major structural changes during the transitions were observed only in the apical domains and the CTSs (Fig. S5 and 6), consistent with both being functionally important and, in agreement with the hypotheses proposed earlier[29], potentially linked.

To investigate the extent of the functional synchrony between the *en bloc* movements of the apical domains and the dynamics of the CTSs, correlations between the conformational paths taken up by these two segments were mapped using an elastic network model, PATH-ENM[49]. This showed that during the transition, the GroEL heptamer traverses through the known nucleotide-driven conformational intermediate states such as the *R* and *R'* states (Fig. 3c and Movie S1). Similarly, the CTSs visited several conformational states, from the rim to the base of the cavity, (Fig. 3b), which triggered gradual opening of the aperture at the bottom of the cavity (Fig. 3c). These observations suggest that the two cavities are partitioned by the CTSs. Interestingly, in agreement with the known motions of the client-binding apical domains, the CTSs exhibited a positively correlated set of large and dynamic conformational transitions with an overall 70° rotation and 45 Å transition, although the constituent equatorial domains remained largely immobile (Fig. 3d and Movie S1). Further, the majority of the fluctuations in the torsion angles φ, ψ and α, and the movement of α carbon atoms calculated at every transition from the *T* state, were observed mostly in the apical domains and the CTSs regions (Fig. 3e), with a perfect anti-correlated motion being observed between these two segments (Fig. S7b), suggesting that they move into the cavity while GroEL moves from the *T* to *R"* state and vice versa (Movie S1). Although the helices H, I, F and M showed characteristic *en bloc* movements while they traversed large molecular distances, the CTSs exhibited dramatic movements as observed by the large fluctuations in all its torsion angles (Fig. 3e). Taken together, these observations support a mechanism where the CTSs exhibit large fluctuations in position that are correlated with the movements of the apical domains, consistent with the hypothesis

that these two segments function in synchrony in client recognition and binding.

### Hydrophobicity and flexibility are critical in *E. coli* GroEL's Gly-Met rich C-terminus

As dynamic fluctuations appear to be a hallmark of the CTS from the analysis above, we next looked at the impact of altering the flexibility of the C-terminal regions. This was done both in silico and experimentally by replacing the glycine residues in the CTS with either alanine or proline. We also examined the effect of altering the hydrophobicity of the CTS, as this may have a role in the potential interactions with client proteins[50,51]. This was done by replacing all the methionine residues with aspartic acid. The resulting GroEL CTS variants, with (AAM)[4]M, (PPM)[4]M, or (GGD)[4]D CTSs, were modelled and subjected to NMA, as described above for the wild-type GroEL. The dynamics of the full-length proteins were largely similar to the wildtype molecules (Fig. S8). However, correlated dynamics of the CTS regions in these variants showed substantial deviations from the wildtype (Fig. 4a). Interestingly the three variants showed distinct dynamics in agreement with their flexibility and charge (Fig. 4a). The wildtype (GGM)[4]M and (GGD)[4]D tails, being flexible sequences, showed diffused dynamic patterns. However, the negative charge introduced by the aspartic acid in (GGD)[4]D, unlike the hydrophobic attraction in the wildtype (GGM)[4]M, appears to have resulted in repulsion within the carboxy termini, which is reflected in the increased dynamics and correlation. The alanine in (AAM)[4]M tails appears to have brought in rigidity to the structure, which is reflected in losing the positive coordination between the subunits. Likewise, the (PPM)[4]M showed higher rigidity and lower positive correlation (Fig. 4a).

Since alteration in the dynamic motions could be functionally significant, we engineered each CTS variant into GroEL and tested the resultant GroEL CTS variants for their ability to replace *groEL* in *E. coli* LG6 and *E. coli* MGM100. None of the three GroEL CTS variants rescued *E. coli* MGM100 under restrictive conditions, while all the variants rescued *E. coli* LG6 (Fig. 4b). This is the same phenotype as is displayed by GroELΔC[13] and suggests that variations in the CTS that alter either hydrophobicity or flexibility have a similar effect on chaperonin function as that caused by complete loss of the CTS. This supports the hypothesis that the composition of CTS is critical to its function, consistent with its high conservation in house-keeping chaperonins.

In summary, the C-terminal segments were observed to assume cavity-specific conformations correlated to the apical domain conformations. This would enable these hydrophobic and structurally flexible CTS to bind and internalize the partially folded (molten globule-like) client proteins via their exposed hydrophobic surfaces (Fig. 5), in agreement with other observations[18,19]. Variations in the CTS that alter flexibility or hydrophobicity adversely affected their role in chaperonin function, suggesting the importance of these features in the C-terminus.

### Discussion

GroEL-GroES mediated protein folding generally requires the formation of a tetradecameric double ring assembly that encloses two isologous cavities for encapsulating client proteins[1,4,38], although some chaperonins can function as heptameric single rings[52–54]. Although the structural changes and functional contributions of the apical, intermediate and the equatorial domains to the chaperonin cycle have been established, the precise role of CTS remains contentious. Previous work suggested that the C-terminal twenty-seven amino acids were dispensable[23–25]. However, the extremely high conservation of the CTS[3,12,33], together with recent in vitro data suggesting an important role in client protein encapsulation and possibly unfolding[18,28,29], led us to re-examine this question using a combination of genetic and structural approaches.

Moreover, paralogues with atypical CTSs have often been demonstrated to be functionally distinct, adopting organism-specific roles which in some cases contribute to pathogenicity[3,22,55–57]. This suggests that the gly-met rich CTS may be a feature of chaperonins with a broader client range and a

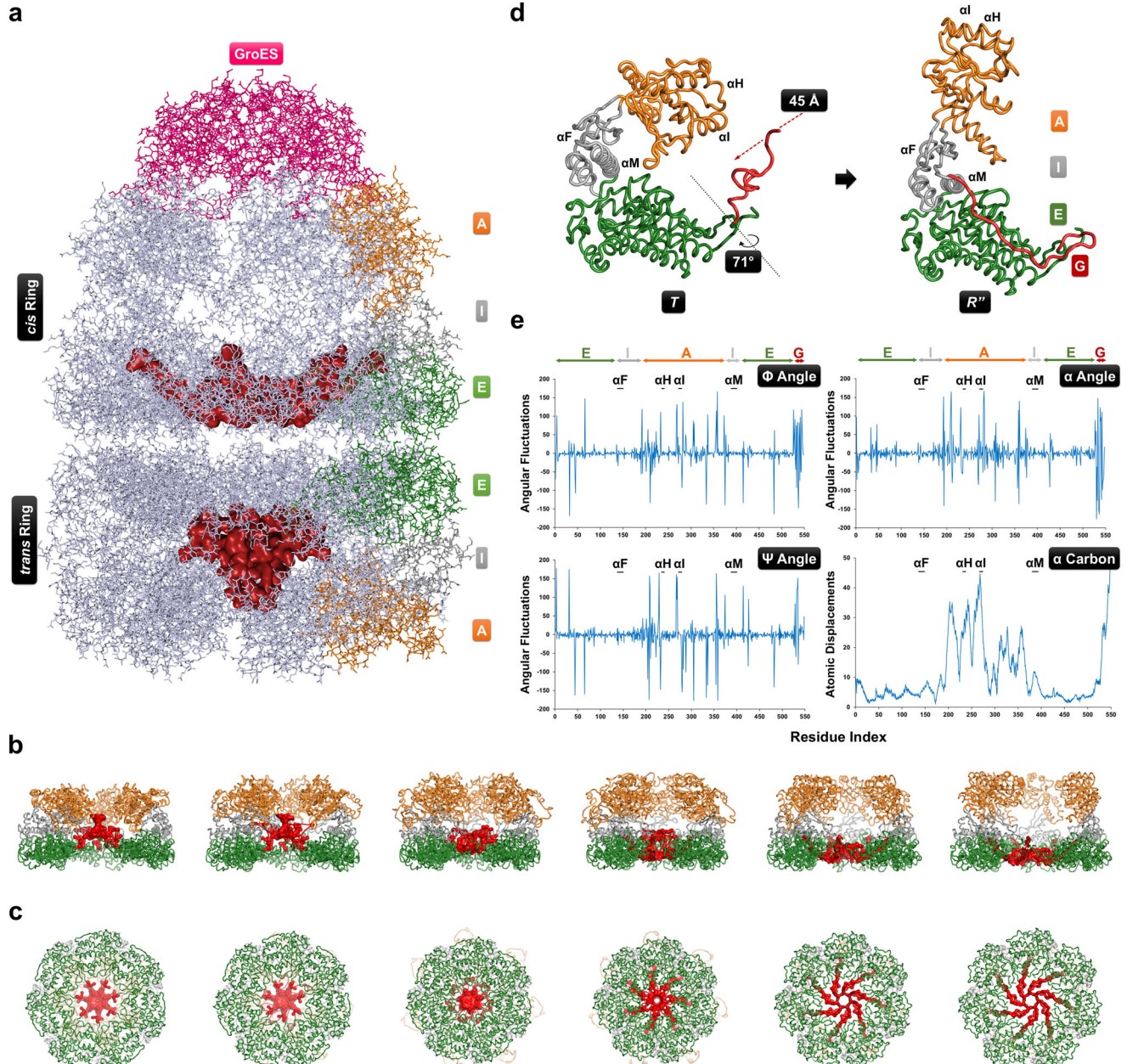

**Fig. 3 | Flexible carboxy termini wobble between the cavity-specific tight (*T*) and relaxed (*R"*) conformations. a** Molecular model of asymmetric GroEL-GroES complexes showing filling of the void by CTSs (red space filled). GroEL and GroES are in pale-blue and pink, respectively. One GroEL subunit in each ring is colour-coded to make the change in domain architecture easier to visualise. A, I and E represent apical, intermediate, and equatorial domains, respectively.
**b** Conformational snapshots showing the series of conformations visited by the GroEL heptamer during transition from the *T* to the *R"* state. Domains in all the subunits are colour-coded as in (**a**). Two subunits were removed in the display to reveal the dynamics of CTSs inside the cavity. **c** Bottom view of the cavity showing the gradual opening of the aperture during the transition, and the dynamic

movement of the CTSs. **d** Cartoon representations of single subunits from the two heptameric rings of GroEL in *T* and *R"* conformational states that were subjected to NMA. The rotation and transition of the CTSs are indicated. The helices F, H, I and M are indicated as αF, αH, αI and αM, respectively. **e** Residue level fluctuations in the torsion angles and displacements of alpha carbon atoms. Fluctuations in the indicated angles and displacements with respect to the *T* state that were calculated for all seven subunits were averaged and plotted as a function of the primary structure of a subunit. E, I, A, and G represent the regions of GroEL primary structure spanning the corresponding domains as colour-coded in the molecular model. The bold lines mark the regions and are scaled according to the size of the indicated helices.

general function in protein folding. These chaperonins therefore appeared to have encountered tighter selection pressure in our phylogenetic analysis (Fig. 1). However, the chaperonins which have different C-termini are likely to have evolved to have more specialized roles[2,3,56]. Interestingly, although the Mycobacterial chaperonins having His-rich CTS have been shown to exist as lower oligomers[58] and thus could not functionally replace *E. coli* GroEL[59], a mutant lacking 18 C terminal residues has demonstrated enhanced ability to assemble into higher order oligomers[60], suggesting that

the His-rich CTS is evolved to provide a specialized or temporal oligo-merization. Further, the observation that the chaperonins with pattern-free CTS are encoded by rapidly growing actinobacteria (*Amycolatopsis nivea*, *M. chubuense*, *M. smegmatis* etc.)[2,3,56], suggests a potential correlation between chaperonin overproduction and growth rate, as was observed in *E. coli*[61]. We further investigated the functional significance of the gly-met-rich CTS in the essential chaperonins using genetic, structural, and computa-tional tools[61].

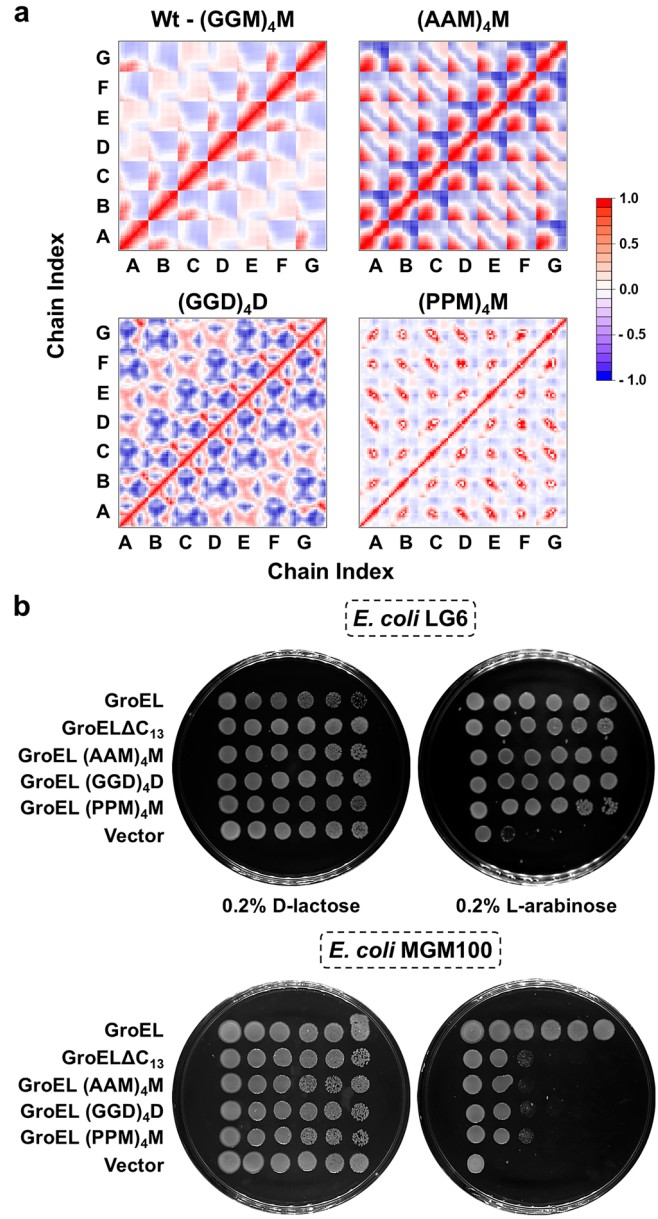

**Fig. 4 | Variations in GroEL carboxy terminus affect chaperonin function.**
**a** Pairwise correlation matrices showing differential displacement of the indicated 13 residue carboxy-terminal peptides of the GroEL CTS variants that are shown and colour-coded as in Fig. S6. A to G indicate the subunit chains in the heptameric ring. **b** Serially diluted cultures of *groEL* conditional mutant strains, *E. coli* LG6 and *E. coli* MGM100 that are expressing the indicated *groEL* CTS variants, were spotted on LB agar plates. The plates were supplemented as indicated and incubated at 30 °C.

We found that GroELΔC13 could complement two *Ts groEL* mutant strains and allowed growth of a strain where the chromosomal *groESL* operon is switched off but expressed from a relatively leaky promoter ($P_{lac}$ in *E. coli* LG6). However, GroELΔC13 could not complement the loss of *groEL* in a strain where either repression of *groESL* was tight (MGM100) or *groEL* was knocked out (AI90) (Fig. 2). Despite numerous attempts we were unable to delete the *groEL* gene in strains expressing GroELΔC13. These observations suggest that at least the last 13 amino acids ((GGM)4M) are necessary for GroEL function. The reason for the discrepancy between these findings and those in some earlier studies on the in vivo activity of C-terminally truncated chaperonins[23–25] is currently not understood, though it may relate to the different strain's growth characteristics[23], levels of chaperonin expression/activity[23], or other experimental conditions used in the previous

studies[24,25]. Given the significant contribution that the CTTs may make to the chaperonin mechanism, as demonstrated both in this paper and in earlier in vitro work[17,19,28,29], this is an area that merits further study. As the expression of GroELΔC13 restored growth of LG6 and complemented both *groEL Ts* mutants, we suggest that mixed GroEL oligomers where a small proportion of the monomers lack the (GGM)4M CTS can function as chaperonins, but those where most or the entire GroEL oligomer is made up of GroELΔC13, do not.

Our structural models showed cavity-specific conformations of CTSs (Fig. 3a) that correlated with the aperture sizes at the base of their respective cavities (Figure S2). Therefore, we speculate that while the smaller aperture of the *trans* cavity may constrain and compel CTSs to protrude into the cavity, the larger aperture at the *cis* cavity provides space for them to adopt a more "flattened" conformation. This is consistent with the observations in the molecular dynamic simulations[32] and cryo-EM studies[28] that CTSs reach the rim of the cavity, where they might interact with the client proteins. The tendency of CTSs to dynamically associate in the *trans* cavity could be enhanced by (i) strong hydrophobic attraction among the CTSs, (ii) repulsion from the positively charged hydrophilic cavity and (iii) space constraints within the trans cavity (Fig. 3). Further, this association is in agreement with the proposed role of the CTS in the assembly of the GroEL tetradecamer[24]. The movement of the hydrophobic CTSs towards the opening of the client-capturing *trans* cavity and segregation in the client-encapsulating *cis* cavity, (Fig. 3c, S7b and Movie S1), suggests a functional linkage between the two CTS conformations and the synchrony with the *en bloc* movements of the client-interacting apical domains. The fact that mutating the CTSs in ways that are expected to alter their flexibility and degree of self-association resulted in the loss of chaperonin function (Fig. 4) further supports these proposed roles of the CTSs in in chaperonin function. Notably, these chaperonin constructs exhibited comparable properties (Fig. S6), indicating the observed loss-of function is due to the altered CTSs. The presence of charged residue-rich CTS in some extremophilic chaperonins[62] and in organisms with multiple chaperonins[3,36], is consistent with the hypothesis that those chaperonin homologues might have functionally diverged to take on the organism-specific roles.

Taken together, employing genetic and structural tools, we have demonstrated that the CTS plays an essential role in the function of GroEL, probably by recognizing the client and/or internalizing it (Movie S1). We therefore propose a model for the CTS's mechanism of action (Fig. 5). In this model, the CTSs bind the unfolded clients with exposed hydrophobic patches. Upon ATP-induced conformational changes of the GroEL ring, these clients are pulled into the cavity by the flattening CTS. Clients can then fold in the hydrophilic cavity during ATP hydrolysis. Following further conformational changes upon ATP binding to the *trans* ring, the folded client that now has a polar surface is ejected from the *cis* ring due to repulsion from the hydrophobic CTSs that push the client outside the cavity. As the CTS is the major divergent sequence feature among the chaperonins in the organisms with multiple chaperonins that exhibit different client pools, it would be interesting to probe if the CTS plays a role in client selection.

As chaperones are involved in several pathogenic processes and implicated in some diseases (the "chaperonopathies"[63,64]), chaperone mechanisms and client repertoire are being explored in developing novel treatment options. For example, selective inhibition of Hsp90 complex assembly and Hsp90-oncogenic client interactions were effective as antitumour agents in vitro and in vivo; several peptidomimetic inhibitors of Hsp90-client interactions are already in various stages of the drug development process[65,66]. Similarly, chaperonin-based therapies are being explored as the stability[67] and folding activity[68] of human chaperonins have been implicated in carcinogenesis. As some of the chaperonins with diverged CTSs have been implicated in pathogenesis, understanding their mechanism of action has the potential to identify new treatment options against the diseases, such as by specific inhibition using CTS-specific nanobodies or siRNA.

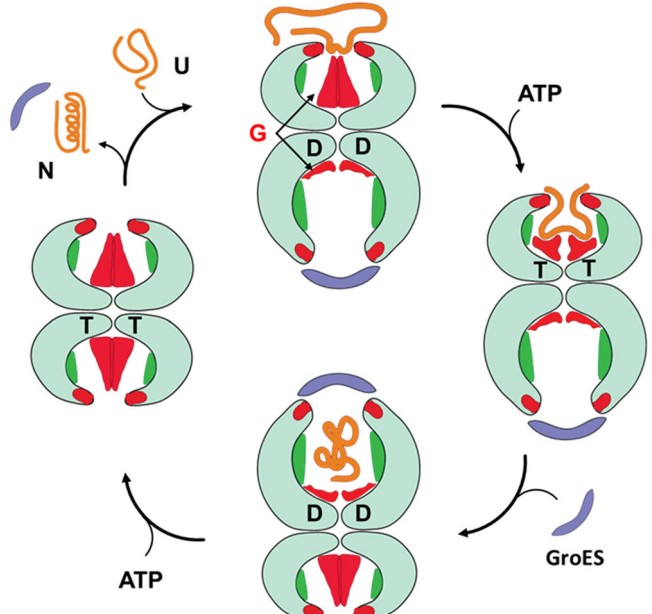

**Fig. 5 | The possible mode of action of carboxy terminal segments.** Model depicting a hypothetical role for CTSs role in the chaperonin mechanism. Please refer to text for details.

## Materials and Methods

### Materials, bacterial strains, and growth conditions

Molecular biology procedures employed in this study were performed according to standard protocols[69]. All chemicals, enzymes and antibiotics were purchased from Sigma Inc. *E. coli* was cultured in standard LB broth supplemented as appropriate. The strains and plasmid vectors employed in this study are listed in Table 1. Briefly, in the *groEL* deletion strain, *E. coli* AI90, the chromosomal *groEL* gene is replaced with a $kan^R$ cassette and GroEL is supplied under lactose inducible $P_{lac}$ promoter control[42] on a p15A, $cam^R$ shelter plasmid, pTGroEL7. GroEL depletion strains *E. coli* LG6[14] and *E. coli* MGM100[43] are derivatives of MG1655 wherein the chromosomal *groESL* operon is placed downstream of $P_{lac}$ and L-arabinose inducible $P_{BAD}$ promoters, respectively. The GroEL *Ts* mutant strains, *E. coli* SV2[45] and *E. coli* A7579[46] are derivative of *E. coli* K12 strains B178 (*galE groESL*⁺) and C600, and bear *Ts* alleles of *groEL*, namely, *groEL44* and *groEL100*, that encode GroEL with point mutations at E191G and S201F, respectively. Plasmids pBAD24[70] and pTrc99A[71] were sourced from lab stocks.

### Phylogenetic analysis

A total of 325 actinobacterial chaperonin paralogue sequences were retrieved from InterPro and KEGG databases and the retrieved entries were pruned to remove repetitions. The amino acid sequences were aligned using MUSCLE alignment programme[72] by 500 iterations of neighbour joining (NJ) algorithm. The alignment was scored with the gap penalties of 2.9 and hydrophobicity multiplied at 1.2. A highly conserved Proline near the C-terminus was identified (position 525 in *E. coli* GroEL) and the amino acid sequence after the proline were considered as CTS. A separate CTS alignment was generated as above. Evolutionary divergence between full-length and CTS sequences was modelled as the rate variation among sites using a gamma distribution with a shape parameter of one. Analyses were conducted using the Poisson correction model[73]. All ambiguous positions were removed for each sequence pair (pairwise deletion option). A total of 639 substitutions in full-length sequences and 57 substitutions in the CTS sequences were observed. The full-length multiple sequence alignment was used in ConSurf to predict the CTS divergence and specific conserved regions among actinobacterial chaperonins. The evolutionary relations

among the aligned sequences and a phylogenetic tree were inferred using the NJ algorithm with 500 rounds of boot strapping MEGA11[74]. The branches belonging to different CTS types were coloured using FigTree v1.4.4. The conservation profiles were mapped onto GroEL structures. The Isoelectric points (pI) and Hydrophobicity (GRAVY) were retrieved from ProtParam tool[75] for each CTS sequence and their distribution was compared with the branching in the phylogenetic trees.

### Generation of GroEL variants

Oligonucleotide primers employed in this study are listed in Supplementary table 1. The plasmid vectors pBAD/GSL[61] and pTrc/GSL[61] harbour the *E. coli* groES-groEL operon cloned into pBAD24[70] and pTrc99A[71]. An opal stop-codon was introduced before 13 residues from the C-terminus in pBAD/GSL and pTrc/GSL using primers GroEL Stop F and GroEL Stop R, resulting in the plasmids pBAD/GroELΔC₁₃ and pTrc/GroELΔC₁₃, respectively, that encode a GroEL variant lacking (GGM)₄M CTS (GroELΔC₁₃). Likewise, constructs expressing GroELΔC₁₆ and GroELΔC₂₈ were generated by introducing opal stop codons into the corresponding codons on pTrc/GSL. GroEL CTS variant clones were generated from pBAD/GSL and pTrc/GSL using synthetic oligonucleotide pairs, AAMF/AAMR, GGDF/GGDR and PPMF/PPMR, as reported to get the pBAD24- and pTrc99a-based GroEL CTS variant clones (Table 1). All the resultant plasmid clones were confirmed by restriction digestion and sequencing.

### Complementation studies

The ability of the GroEL carboxyl terminus variants to functionally replace *E. coli* GroEL was assessed in *E. coli* groEL conditional mutant strains LG6[14,42], essentially as reported earlier[59] and MGM100[43], *Ts* mutant strains, SV2[45] and A7579[46], and deletion stain AI90[42], essentially as reported earlier[59]. Briefly, *E. coli* LG6, wherein the chromosomal *groESL* operon is under the control of $P_{lac}$, was transformed with the pBAD24-derived clones of the *groEL* CTS variants, and the activity of the cloned genes was scored in the presence of 0.2% L-arabinose, while *E. coli* MGM100, wherein the chromosomal copy of *groE* is under the control of $P_{BAD}$ was transformed with the pTrc99A derived clones of *groEL* CTS variants, and their activity was scored in the presence of 0.2% D-lactose. Plates with either 0.2% D-lactose or 0.2% L-arabinose were included as positive controls, as these are permissive conditions for the strains *E. coli* LG6 and *E. coli* MGM100, respectively. For the complementation studies with *groEL ts* mutant strains, the strains *E. coli* SV2 and *E. coli* A7579 were transformed with pBAD24 derived clones of *groEL* CTS variants (Table 1) and the cultures expressing GroEL variants were spotted onto two LB agar plates supplemented with 0.2% L-arabinose. One of the plates was incubated at 30 °C and the other was incubated at 42 °C, representing permissive and restrictive conditions, respectively. Likewise, *E. coli* AI90 was transformed with plasmids expressing wildtype *groEL* pTrc/GSL, *groEL* CTS variants, pTrc/GroELΔC₁₃, pTrc/GroELΔC₁₆ and pTrc/GroELΔC₂₈, and the vector control, pTrc99A. The resultant transformants were cultured in LB supplemented with Amp, Kan, and 0.2% D-lactose, serially diluted and spotted onto three sets of LB agar plates, the first set supplemented with Amp, Kan, and 0.2% D-lactose and the second set supplemented with Amp, Kan, Cam, and 0.2% D-lactose. The plates were incubated at 30, 37, and 42 °C.

### Reverse transcriptase assay PCR to assess expression levels of GroEL

A semi-quantitative expression assay was employed to assess the expression levels of chromosomal and vector-borne *groEL* genes in *E. coli* MGM100 and *E. coli* LG6. For this, *E. coli* MGM100 and *E. coli* LG6 were transformed with pTrc/GSL and pBAD/GSL, respectively, and were cultured in LB supplemented with 0.2% D-lactose and 0.2% L-arabinose, respectively, to induce the vector-borne *groEL*. The cultures were recovered in late-log phase, and the total RNA was isolated using TRIzol reagent extraction method. 1 μg of total RNA from each culture was reverse

**Table 1 | *E. coli* Strains and Plasmid Vectors used in this Study**

| Name | Description | Source or Reference |
|---|---|---|
| *Strains* | | |
| *E. coli* AI90 | *groEL* deletion strain - *groEL* on a p15A cam$^R$ shelter plasmid is regulated by $P_{lac}$ promoter | 42 |
| *E. coli* LG6 | Chromosomal *groES/L* operon under lactose/IPTG inducible $P_{lac}$ promoter | 14 |
| *E. coli* MGM100 | Chromosomal *groES/L* operon under arabinose inducible $P_{BAD}$ promoter | 43 |
| *E. coli* SV2 | B178 derived temperature sensitive strain having *groEL44* allele encoding GroEL E191G | 45 |
| *E. coli* A7579 | C600 derived temperature sensitive strain having *groEL100* allele encoding GroEL S201F | 46 |
| *Plasmid Vectors* | | |
| pBAD24 | L-arabinose inducible expression vector, ColE1 origin, Amp$^R$. | 70 |
| pTrc99A | $P_{tac}$ based expression vector, ColE1 origin Amp$^R$. | 71 |
| pBAD/GSL | *E. coli groESL* operon cloned in pBAD24 | 61 |
| pTrc/GSL | *E. coli groESL* operon cloned in pTrc99A | 61 |
| pBAD/GroEL$\Delta$C$_{13}$ | pBAD24 harbouring the ORF for GroEL$\Delta$C$_{13}$. | This study |
| pBAD/AAM | pBAD24 harbouring the ORF for *E. coli* GroEL with (AAM)$_4$M C-terminus | This study |
| pBAD/GGD | pBAD24 harbouring the ORF for *E. coli* GroEL with (GGD)$_4$D C-terminus | This study |
| pBAD/PPM | pBAD24 harbouring the ORF for *E. coli* GroEL with (PPM)$_4$M C-terminus | This study |
| pTrc/GroEL$\Delta$C$_{13}$ | pTrc99a harbouring ORF for GroEL$\Delta$C$_{13}$ | This study |
| pTrc/AAM | pTrc99a harbouring the ORF for *E. coli* GroEL with (AAM)$_4$M C-terminus | This study |
| pTrc/GGD | pTrc99a harbouring the ORF for *E. coli* GroEL with (GGD)$_4$D C-terminus | This study |
| pTrc/PPM | pTrc99a harbouring the ORF for *E. coli* GroEL with (PPM)$_4$M C-terminus | This study |

transcribed with MMLV reverse transcriptase, using vector-specific primers that bind downstream of the Multiple Cloning Site on the vectors; pTrc R for MGM100 + pTrc/GSL culture and pBAD R for LG6 + pBAD/GSL cultures. These primers, therefore, specifically reverse transcribe the vector-borne mRNA but not the chromosome-borne mRNA. For the amplification of the resulting cDNA, a forward primer GroELRTF was employed, which binds within *groEL* gene upstream of the C-terminus and with the corresponding reverse primers, either pTrc R or pBAD R, results in a product of about 300 bp, the intensity of which is proportional to the levels of corresponding mRNA. The PCR-amplified products were resolved on 3% agarose gel.

### Homology modelling of *E. coli* GroEL Caboxy terminal sequence region

Since the crystal structures of *E. coli* GroEL lack the CTSs[5,6], we have modelled these regions into *E. coli* GroEL. Three-dimensional co-ordinates for *E. coli* GroEL subunits from the *cis* and the *trans* cavities, which represent relaxed ($R$") and tight ($T$) conformational states, respectively, were sourced from the crystal structure, PDB ID: 1AON[6]. CTSs were modelled into these subunit monomers using Modeller 9.14[76] and short contaCTS were corrected using the Crystallographic Object-Oriented Toolkit (*Coot*) 0.8[77]. The models with low Root Mean Square Deviation (RMSD) were selected, superimposed onto the co-ordinates of *E. coli* GroEL using the iSuperpose application hosted at Mobyle portal at Ressource Parisienne en BioInformatique Structurale (RPBS), to obtain symmetric and asymmetric tetradecamer models of GroEL with CTS and the resulting models of GroEL tetradecamers were used for further analysis.

### Normal mode analysis of the GroEL rings

To understand the transitions of the CTSs between the two conformations, Normal Mode Analysis (NMA) with PATH-ENM was performed on the GroEL protomer and heptamer models, in $T$ and $R$" states, following the methods reported earlier[49,78,79]. PATH-ENM built two ENM potentials for each reference structure and combined them into an interpolated mixed potential. At each transition, therefore, the mixed potential had two minima: one for each structure and one saddle point representing the transition state. Ultimately, the transition paths between

two end structures were generated using this Mixed Elastic Network Model (MENM). Transitions of the CTSs between the two end states were generated by calculating the contribution of each normal mode to the observed conformational change using Elastic Network Model[80,81]. The individual normal modes and their deformations were visualized using Pymol 1.3. Additionally, NMA using Molecular Modelling Toolkit (MMTK) package with C-alpha force field was performed on the GroEL protomer and heptamers. In each case, modes with the least deformation energy and eigenvalues were considered for calculating atomic displacements and for further analysis. Molecular fluctuations and correlated movement of the alpha carbons in the molecule were calculated following covariance analysis[82–84].

### Reporting summary
Further information on research design is available in the Nature Portfolio Reporting Summary linked to this article.

## Data availability
All material and other data are available from the corresponding author upon reasonable request. The plasmids generated in this study will be submitted to Addgene at our lab's page - https://www.addgene.org/plasmids/articles/28253031/.

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

## Acknowledgements

We thank Arthur Horwich, Millicent Masters, Alan Fersht, Costa Georgopoulos, and Ariella Oppenheim for the GroE mutant strains, *E. coli* LG6, MGM100, SV2 (*groEL44*), and A7579 (*groEL100*), respectively. We thank Chris Bunse, Damon Huber, Jayaraman Gowrishankar, Jayant Udgaonkar, Sandhya Visweswaraiah, and Abhijit A. Sardesai for helpful discussions. This research was supported by the grants from the Biotechnology and Biological Sciences Research Council (BBSRC), UK (BB/S017526/1), CMSK's Royal Society Newton International Fellowship (NF161469), and the Department of Biotechnology, India (BT/PR3260/BRB/10/967/2011).

## Author contributions

C.M.S.K., S.C.M., and P.A.L. conceptualized the idea. C.M.S.K. conducted the experiments and A.M.M. assisted in Phylogenetic analysis. C.M.S.K., S.C.M. and P.A.L. supervised the investigation, discussed the findings, and written the manuscript.

## Competing interests

The authors declare no competing interests.
