## [Transparent Peer Review file · Communications Biology]

Genetic and Structural insights into the Functional Importance of the Conserved Gly-Met-Rich C-Terminal Tails in Bacterial Chaperonins

Corresponding Author: Dr Santosh Kumar

Version 0:

Reviewer comments:

Reviewer #1

(Remarks to the Author)
Kumar et al.

The C-terminal tails of GroEL subunits are established to be important for chaperonin function. Due to their dynamic nature, the C-tails have been challenging to characterise structurally, and their precise role in the GroEL/ES functional cycle is relatively poorly understood. In this manuscript, Kumar et al. use genetics and structural modelling to study the function and conformational ensemble of GroEL C-tails. They find that the length and sequence composition of the C-tails are critical for GroEL function in *E. coli*, and describe coordinated dynamics of the C-tails and apical domains. On the basis of these and previous findings, they propose a model for the functional cycle of GroEL that emphasizes the C-tails.

This is a nice, clearly written paper that makes a useful contribution to our understanding of chaperonin function. I have only minor comments.

1. Line 26. The sentence beginning "Our findings clear the conundrum..." is in my opinion overstated. I suggest removing this sentence, or making a weaker claim.

2. Line 48. "The AlphaFold predicted models for *E. coli* (<https://alphafold.ebi.ac.uk/entry/P0A6F5>) and *Bacillus subtilis* (<https://alphafold.ebi.ac.uk/entry/P28598>) show extended carboxy termini as the models only considered monomeric forms, not the energy-efficient oligomeric single rings or bullets."

I'm not sure exactly what is meant here. Is the implication that AlphaFold would model the termini differently in the oligomer? Is there any evidence for this? Can the authors simply test this by modelling a GroEL ring using AlphaFold 3?

3. Figure 4a. NMA of C-tail variants. Can the authors comment on why the GGD variant is so different to the WT GGM in terms of dynamics?

4. Line 739. Fig. 5. Although the present study shows that the C-tails are essential, and previous work has demonstrated a role in client recognition and exit, it does not follow that recognition/exit are the essential functions of the C-tails (as opposed to e.g. an effect on in-cage folding). The model is very nice, but I suggest rephrasing the figure title.

David Balchin

Reviewer #2

(Remarks to the Author)

This study follows up on extensive and solid work by this group in which the existence and putative functions of multicopy GroEL paralogs in bacterial species have been explored. The main body of extant molecular work on GroEL has focused on the *E. coli* system but the extraordinary range of isoforms of GroEL has been asking for mechanistic analysis to assign functions to the less conserved domains of GroEL. The current ambitious work attempts to place the C terminal segments (CTSs) into context with different consortia of GroEL paralogs. The multiple alignments show that while one class of paralogs has conserved Gly-Met-rich tails, non-canonical classes exists with relatively unconserved CTS sequence and composition. Thus, it is a complex undertaking to sort this out. On the one hand, in the *E. coli* GroEL system the CTS has been the target for multiple structure/function studies that reveal that it is partially dispensable for many classes of client proteins, and for assembly of the cage. The experimental approach of this paper relies heavily on the functional properties of the 13 residue C-terminal deletion of *E. coli* GroES. The findings are in disagreement with three studies (refs 23-25) that

appear to establish that large C-terminal deletions DO have functional tetradecamers. In reference 24, Burnett et al 1994) establish that up to 27 residues of the C-terminus of E. coli GroES could be deleted "without any observable phenotypic effect." (Discussion, line 5). References 12 and 13 describe work with amplified tail variants of E. coli GroEL and would establish that the gross properties of the CTS have to be altered in order to affect folding dynamics. The CTS could be tripled in size without much effect (except on the largest folding substrate, Rubisco) and the 4X repeated CTS did have a marked inhibitory effect as well as curtailing in vivo effectiveness. The working model shown in Fig 5 has a CTS deleted strain depicted as non-functional which is hard to reconcile with these earlier studies.

This paper thus tackles a problem in assigning putative client recognition functions to a class of CTS sequences that have really extensive redundancy and robust structural features that continue to function despite quite radical mutational alteration or divergence. There are some statements that must be clarified to have provide a measure of the confidence that can be assigned to assumptions implicit in the Discussion. For example, in the first line of the discussion, "GroEL-GroES mediated protein folding requires the formation of a tetradameric double ring assembly that encloses two isologous cavities for encapsulating client proteins." fails to recognized a large body of work proving that single ring variants of E. coli GroEL-ES are functional in vivo and in vitro. Also, Ref 55 points out that Mycobacterial GroEL may have chaperone activities when they are in lower order oligomers than the cage complexes.

The approach pioneered here, namely identifying classes of mutants with modulation of the client protein specific binding of the GroEL has great potential for identifying the mechanistic basis for sorting client proteins from non-client proteins. The enigmatic sorting interaction involves unfolded proteins of presumably amorphous structure. How this works is a very important open question in this otherwise intensely studied field and is an important research frontier.

Reviewer #3

(Remarks to the Author)

This is an interesting paper on the investigation of the role of the conserved glycine and methionine-rich carboxy-terminal residues in chaperonine GroEL in bacterial models. The authoes show via structural modelling that the C-terminal region shuttles between two cavity-specific conformations that correlate with the client-protein-binding apical domains, supporting C-termini's role in client protein encapsulation. They expand their research to demonstrate that the gly-met-rich C-termini are functionally significant in chaperonin-mediated protein folding function using phylogenetic analysis and sequence tools. The paper is interesting and worth publications.

To make their paper of even larger impact the authors could discuss the implication of their work for therapeutic applications, discussing for instance at what was done in this and other chaperones (10.1002/chem.202000615; 10.1038/s41594-024-01352-0; 0.1016/j.apsb.2020.11.015) with regards to client protein folding and treatment options.

Version 1:

Reviewer comments:

Reviewer #2

(Remarks to the Author)

The revised manuscript COMMSBIO-24-7228A has been significantly improved since the original submission.

This study seeks to define the specialized functions of the C-terminal segments of Type 1 Cpn60 complexes. These highly conserved complexes often exist as multiple paralogs in single strains (for example in *Bradyrhizobium japonicum* encoding seven GroEL homologs). As background to this, the most thoroughly studied GroEL from E. coli has been shown to function, albeit weakly, with the terminal 27 amino acids deleted. In bacteria with multiple homologs, the variability of sequence conservation between homologs is concentrated in the C terminal segments, suggesting that this region might be a major determinant in controlling the roles of these coexpressed paralogs. In this reviewer's opinion, this paper breaks new ground in discerning the relative roles of multiple GroEL homologs as well as in examining the contributions of the C terminal segments to fitness in organisms with a single homolog.

This paper, with revisions, represents a big step in the right direction in resolving the debate about the function of the Cpn60 C-terminus which the authors correctly label as contentious. I therefore strongly support acceptance. The addition of a well supported model for potentially sorting client proteins from the huge array of non-client proteins will result in significant downloads and citations in a field that has not seen significant advances in the last 5 years.

Point-by-point response to the reviewer's comments.

Reviewers' comments:

Reviewer #1 (Remarks to the Author):

Comment: Kumar et al.

The C-terminal tails of GroEL subunits are established to be important for chaperonin function. Due to their dynamic nature, the C-tails have been challenging to characterise structurally, and their precise role in the GroEL/ES functional cycle is relatively poorly understood. In this manuscript, Kumar et al. use genetics and structural modelling to study the function and conformational ensemble of GroEL C-tails. They find that the length and sequence composition of the C-tails are critical for GroEL function in *E. coli*, and describe coordinated dynamics of the C-tails and apical domains. On the basis of these and previous findings, they propose a model for the functional cycle of GroEL that emphasizes the C-tails.

This is a nice, clearly written paper that makes a useful contribution to our understanding of chaperonin function. I have only minor comments.

Response: We thank the reviewer for appreciating our work.

Comment 1. Line 26. The sentence beginning “Our findings clear the conundrum...” is in my opinion overstated. I suggest removing this sentence, or making a weaker claim.

Response: We acknowledge the reviewer for the suggestion. We have removed the sentence.

Comment 2. Line 48. “The AlphaFold predicted models for *E. coli* (<https://alphafold.ebi.ac.uk/entry/P0A6F5>) and *Bacillus subtilis* (<https://alphafold.ebi.ac.uk/entry/P28598>) show extended carboxy termini as the models only considered monomeric forms, not the energy-efficient oligomeric single rings or bullets.” I'm not sure exactly what is meant here. Is the implication that AlphaFold would model the termini differently in the oligomer? Is there any evidence for this? Can the authors simply test this by modelling a GroEL ring using AlphaFold 3?

Response: We thank the reviewer for this suggestion. Apologies for unclear phrasing. We tested this hypothesis as explained below.

Following the reviewer's suggestions, we modelled GroEL heptamer using AlphaFold 3 (<https://alphafoldserver.com/>), using full length GroEL sequence. The resultant heptamer was in the tight conformation (Review Fig. 1A). The prediction was largely accurate as the predicted template modeling (pTM) score was 0.75 (a score above 0.5 is considered reliable, <https://doi.org/10.1093/bioinformatics/btq066>) and the interface predicted template modeling (ipTM) score (measures the relative subunit position accuracy) was 0.72 (close to the high-quality prediction score of 0.8, <https://doi.org/10.1002/prot.20264>).

Although majority of the model showed high confidence, the CTS region showed lower confidence (pIDDT < 50, Review Fig. 1A), as shown in the molecular model (in orange) and the heatmap (clear zones at the ends of subunits). Similar confidence levels (lower confidence for

CTS region) were observed in the monomer's model (<https://alphafold.ebi.ac.uk/entry/P0A6F5>, Review Fig. 1B).

To test our hypothesis on the CTS conformation, we superposed the AlphaFold monomer (<https://alphafold.ebi.ac.uk/entry/P0A6F5>) onto the modelled heptamer. The CTS of the monomer (Review Fig. 1C and 1D, shown in orange) was observed to be extended and reach the protomers on the other side of the heptamer, as shown in the side (Review Fig. 1C) and top (Review Fig. 1D) views. This conformation is likely to cause steric hindrance while forming a heptameric (and tetradecameric) assembly. The superposed protomer in the ring is shown in green, while other six protomers are shown in grey.

Review Fig. 1. Structural comparison of AlphaFold predicted monomeric and heptameric (single ring) GroEL structures.

We could not model a tetradecamer, despite repeated attempts. Similar to our model (Fig. 3), AlphaFold 3 predicted that the CTS reached the rim, in the tight conformation. Interestingly, AlphaFold 3 modelled a helix within the CTS, which was not observed in our model. This raised interesting questions on the packaging of CTS in the tight conformation. We are keen to pursue this further.

Therefore, we revised the sentences (Lines 46 - 50) as below.

Structural models of *E. coli* (<https://alphafold.ebi.ac.uk/entry/P0A6F5>) and *Bacillus subtilis* (<https://alphafold.ebi.ac.uk/entry/P28598>) GroELs generated by the AlphaFold show extended carboxy termini. These models considered monomeric forms, not the energy-efficient oligomeric single ring or bullet forms. The extended conformation of the carboxy termini would cause steric hindrance when monomers assemble into single or double ring conformations.

Comment 3. Figure 4a. NMA of C-tail variants. Can the authors comment on why the GGD variant is so different to the WTGGM in terms of dynamics?

Response: We thank the reviewer for alerting on this. Following the reviewer's suggestion, we revised the sentences (line 254-262) as below.

Interestingly the three variants showed distinct dynamics in agreement with their flexibility and charge (Fig. 4a). The wildtype (GGM)₄M and (GGD)₄D tails, being flexible sequences, showed diffused dynamic patterns. However, the negative charge introduced by the aspartic acid in (GGD)₄D, unlike the hydrophobic attraction in the wildtype (GGM)₄M, appears to have resulted in repulsion within the carboxy termini, which is reflected in the increased dynamics and correlation. The alanine in (AAM)₄M tails appears to have brought in rigidity to the structure, which is reflected in losing the positive coordination between the subunits. Likewise, the (PPM)₄M showed higher rigidity and lower positive correlation (Fig 4a).

Comment 4. Line 739. Fig. 5. Although the present study shows that the C-tails are essential, and previous work has demonstrated a role in client recognition and exit, it does not follow that recognition/exit are the essential functions of the C-tails (as opposed to e.g. an effect on in-cage folding). The model is very nice, but I suggest rephrasing the figure title.

Response: We acknowledge the reviewer for alerting on this. We have modified the title and caption for figure 5, as below.

Figure 5. The possible mode of action of carboxy terminal segments. Model depicting a hypothetical role for CTSs role in the chaperonin mechanism. Please refer to text for details.

Reviewer #2 (Remarks to the Author):

Comment: This study follows up on extensive and solid work by this group in which the existence and putative functions of multicopy GroEL paralogs in bacterial species have been explored. The main body of extant molecular work on GroEL has focused on the E. coli system but the extraordinary range of isoforms of GroEL has been asking for mechanistic analysis to assign functions to the less conserved domains of GroEL. The current ambitious work attempts to place the C terminal segments (CTSs) into context with different consortia of GroEL paralogs. The multiple alignments show that while one class of paralogs has conserved Gly-Met-rich tails, non-canonical classes exists with relatively unconserved CTS sequence and composition. Thus, it is a complex undertaking to sort this out. On the one hand, in the E coli GroEL system the CTS has been the target for multiple structure/function studies that reveal that it is partially dispensable for many classes of client proteins, and for assembly of the cage. The experimental approach of this paper relies heavily on the functional properties of the 13 residue C-terminal deletion of E coli GroES. The findings are in disagreement with three studies (refs 23-25) that appear to establish that large C-terminal deletions DO have functional tetradecamers. In reference 24, Burnett et al 1994) establish that up to 27 residues of the C-terminus of E. coli GroES could be deleted "without any observable phenotypic effect." (Discussion, line 5). References 12 and 13 describe work with amplified tail variants of E. coli GroEL and would establish that the gross properties of the CTS have to be altered in order to affect folding dynamics. The CTS could be tripled in size without much effect (except on the largest folding

substrate, Rubisco) and the 4Xrepeated CTS did have a marked inhibitory effect as well as curtailing in vivo effectiveness.

Response: We acknowledge the reviewer for carefully reviewing our manuscript. We have tested these concerns earlier using different GroEL truncation variants (tail deletion mutants) (Review Fig. 2). As we could not identify a possible reason for the discrepancy, we did not include these observations in the manuscript.

As noted by the reviewer, a variant of GroEL lacking its last sixteen amino acids was reported as being able to act as the only GroEL protein in *E. coli*, allowing growth and plating of various phage that require GroEL for function (McLennan et al. 1993). The only defect reported for this strain was an extended lag time coming out of stationary phase, when cells had been incubated at 42 °C, and the purified protein was shown to exhibit slightly lower ATPase rate than wild type GroEL. Subsequent work also reported that larger C-terminal deletions (up to twenty-seven amino acids) were also functional, although some of these had to be produced at high levels to see maximum function (Burnett et al. 1994). Because of the formal possibility that GroEL Δ_{13} that we investigated might be non-functional, but a larger deletion might restore function, we constructed the same 16 amino-acid deletion that was made by McLennan et al. However, as described below, in our hands this truncation mutant was also not functional.

Briefly, we tested two different GroEL truncation variants, GroEL Δ_{16} and GroEL Δ_{28} , as GroEL Δ_{16} was previously described as being functional, while GroEL Δ_{28} was the first truncation mutant described as being non-functional (McLennan et al. 1993, Burnett et al. 1994, McLennan et al. 1994). All three variants were tested alongside the full-length GroEL for their ability to function in the absence of the chromosome-encoded GroEL. As can be seen in Review Fig. 2, only the full-length protein was able to allow the loss of the shelter plasmid (pTG7) and hence for the strain losing it to become chloramphenicol sensitive.

Review Fig. 2. Complementation of different GroEL truncation variants in Al90 $\Delta groEL$ strain. Different variants were expressed on lactose-inducible pTrc99A, and ten-fold dilutions of each of these strains were spotted on the plates as shown. W: full length wild type GroEL; V: vector only.

As the complementation method involves screening for the loss of chloramphenicol resistance, and hence rare events would be hard to detect, we further attempted to P1 transduce the $\Delta groEL::kan^R$ marker from Al90 into *E. coli* MG1655 harbouring the plasmids encoding the same *groEL* tail variants shown in Review Figure 2. We were unable to obtain any *kan^R* colonies with strains harbouring plasmids expressing any of the truncated GroEL mutants, whereas the strain harbouring the plasmid expressing GroEL_{wt} readily gave rise to *kan^R* colonies at high frequency.

As we were unable to explain the contradiction, we omitted these observations in the manuscript. We suppose that further work is needed to understand the reasons for these contradictory observations.

Following reviewer's concern, we revised the discussion as below (Lines 304 –320):

We found that GroEL Δ C₁₃ could complement two *Ts groEL* mutant strains and allowed growth of a strain where the chromosomal *groESL* operon is switched off but expressed from a relatively leaky promoter (*P_{lac}* in *E. coli* LG6). However, GroEL Δ C₁₃ could not complement the loss of *groEL* in a strain where either repression of *groESL* was tight (MG100) or *groEL* was knocked out (AI90) (Fig. 2). Despite numerous attempts we were unable to delete the *groEL* gene in strains expressing GroEL Δ C₁₃. These observations suggest that at least the last 13 amino acids ((GGM)₄M) are necessary for GroEL function. The reason for the discrepancy between these findings and those in some earlier studies on the in vivo activity of C-terminally truncated chaperonins²³⁻²⁵ is currently not understood, though it may relate to the different strain's growth characteristics²³, levels of chaperonin expression/activity²³, or other experimental conditions used in the previous studies^{24,25}. Given the significant contribution that the CTSs may make to the chaperonin mechanism, as demonstrated both in this paper and in earlier in vitro work^{17,19,28,29}, this is an area that merits further study. As the expression of GroEL Δ C₁₃ restored growth of LG6 and complemented both *groEL Ts* mutants, we suggest that mixed GroEL oligomers where a small proportion of the monomers lack the (GGM)₄M CTS can function as chaperonins, but those where most or the entire GroEL oligomer is made up of GroEL Δ C₁₃, do not.

We hope our explanation addresses the reviewer's concerns.

Comment: The working model shown in Fig 5 has a CTS deleted strain depicted as non-functional which is hard to reconcile with these earlier studies.

Response: We acknowledge the reviewer for carefully reviewing our manuscript. Following our explanation on the discrepancy with previous studies (please see above) and our observations in this study, we propose that the CTS is key in chaperonin function. However, as more work is needed to comprehend the link between CTS and chaperonin function, and following reviewer's concern, we limited Figure 5 to CTSs' mechanism and removed the functional and non-functional section.

Comment: This paper thus tackles a problem in assigning putative client recognition functions to a class of CTS sequences that have really extensive redundancy and robust structural features that continue to function despite quite radical mutational alteration or divergence. There are some statements that must be clarified to have provide a measure of the confidence that can be assigned to assumptions implicit in the Discussion. For example, in the first line of the discussion, "GroEL-GroES mediated protein folding requires the formation of a tetradecameric double ring assembly that encloses two isologous cavities for encapsulating client proteins." fails to recognized a large body of work proving that single ring variants of *E. coli* GroEL-ES are functional in vivo and in vitro.

Response: We thank the reviewer for alerting us on this. In this statement, we were starting the discussion with a statement on the general understanding of the mechanism of action of chaperonins, which is why this sentence did not include a mention of single ring chaperonin variants. However, we agree this is an important point, so we have added a sentence as below (lines 279-281).

GroEL-GroES mediated protein folding generally requires the formation of a tetradecameric double ring assembly that encloses two isologous cavities for encapsulating client proteins^{1,4,38}, although some chaperonins can function as heptameric single rings⁵²⁻⁵⁴.

Comment: Also, Ref 55 points out that Mycobacterial GroEL may have chaperone activities when they are in lower order oligomers than the cage complexes.

Response: We thank the reviewer for pointing out on this. The reference 55 (now 57, Qamra, et al. 2004. *Mycobacterium tuberculosis* GroEL homologues unusually exist as lower oligomers and retain the ability to suppress aggregation of substrate proteins. *JMB*) demonstrates that the *Mycobacterial* chaperonins, when produced recombinantly in *E. coli* fail to assemble as higher order oligomers. Therefore, these recombinant chaperonins failed to exhibit ATPase activity, or refold model client proteins, rhodanase and citrate synthase. However, owing to their hydrophobic apical domains (showed by their bis-ANS fluorescence), these chaperonins effectively bind unfolded client proteins and prevent their aggregation.

The authors also state in reference 57 : “Our results therefore suggest that the *M. tuberculosis* Cpn60s are efficient in preventing aggregation of denatured proteins, but do not possess ATP-dependent chaperoning activity.”

We agree with the statement completely as the third author (SCM) of this manuscript was the senior author on reference 57, the first author (CMSK) was part of the lab when the manuscript was published; we had several discussions on the data. Subsequent published work did show that the Mycobacterial chaperonins have to assemble into oligomers in order to become fully functional in vivo (see for example <https://pubmed.ncbi.nlm.nih.gov/22834700/>). We’ve not cited this work to avoid making the current paper over-complicated.

I hope our explanation addressed the reviewer’s concerns.

Comment: The approach pioneered here, namely identifying classes of mutants with modulation of the client protein specific binding of the GroEL has great potential for identifying the mechanistic basis for sorting client proteins from non-client proteins. The enigmatic sorting interaction involves unfolded proteins of presumably amorphous structure. How this works is a very important open question in this otherwise intensely studied field and is an important research frontier.

Response: We acknowledge the reviewer appreciating our work and direction. We did in fact explore client protein specificity, using quantitative proteomics, and we identified some clients (about 30) that may require the CTS for folding. Interestingly, the strain expressing GroEL Δ C₁₃ showed enhancement of other chaperones and redox proteins. However, as the data needed further work, we omitted them from this manuscript. We are keen to explore this further with different tail variants and different bacterial systems.

Reviewer #3 (Remarks to the Author):

Comment: This is an interesting paper on the investigation of the role of the conserved glycine and methionine-rich carboxy-terminal residues in chaperonin GroEL in bacterial models. The authors show via structural modelling that the C-terminal region shuttles between two cavity-specific conformations that correlate with the client-protein-binding apical domains, supporting C-termini’s role in client protein encapsulation. They expand their research to

demonstrate that the gly-met-rich C-termini are functionally significant in chaperonin-mediated protein folding function using phylogenetic analysis and sequence tools. The paper is interesting and worth publications.

Response: We thank the reviewer for appreciating our work.

Comment: To make their paper of even larger impact the authors could discuss the implication of their work for therapeutic applications, discussing for instance at what was done in this and other chaperones (10.1002/chem.202000615; 10.1038/s41594-024-01352-0; 10.1016/j.apsb.2020.11.015) with regards to client protein folding and treatment options.

Response: We thank the reviewer for this suggestion. Following the suggestion, we revised the discussion (Lines 354-364) as below.

As chaperones are involved in several pathogenic processes and implicated in some diseases (the “chaperonopathies”^{60,61}), chaperone mechanisms and client repertoire are being explored in developing novel treatment options. For example, selective inhibition of Hsp90 complex assembly and Hsp90-oncogenic client interactions were effective as anti-tumour agents *in vitro* and *in vivo*; several peptidomimetic inhibitors of Hsp90-client interactions are already in various stages of the drug development process^{62,63}. Similarly, chaperonin-based therapies are being explored as the stability⁶⁴ and folding activity⁶⁵ of human chaperonins have been implicated in carcinogenesis. As some of the chaperonins with diverged CTSs have been implicated in pathogenesis, understanding their mechanism of action has the potential to identify new treatment options against the diseases, such as by specific inhibition using CTS-specific nanobodies or siRNA.